# Autophagy: A Cellular Guardian against Hepatic Lipotoxicity

**DOI:** 10.3390/genes14030553

**Published:** 2023-02-22

**Authors:** Rohit Anthony Sinha

**Affiliations:** Department of Endocrinology, Sanjay Gandhi Postgraduate Institute of Medical Sciences, Lucknow 226014, India; rasinha@sgpgi.ac.in

**Keywords:** lipotoxicity, NAFLD, NASH, autophagy, ER stress, oxidative stress

## Abstract

Lipotoxicity is a phenomenon of lipid-induced cellular injury in nonadipose tissue. Excess of free saturated fatty acids (SFAs) contributes to hepatic injury in nonalcoholic fatty liver disease (NAFLD), which has been growing at an unprecedented rate in recent years. SFAs and their derivatives such as ceramides and membrane phospholipids have been shown to induce intrahepatic oxidative damage and ER stress. Autophagy represents a cellular housekeeping mechanism to counter the perturbation in organelle function and activation of stress signals within the cell. Several aspects of autophagy, including lipid droplet assembly, lipophagy, mitophagy, redox signaling and ER-phagy, play a critical role in mounting a strong defense against lipotoxic lipid species within the hepatic cells. This review provides a succinct overview of our current understanding of autophagy–lipotoxicity interaction and its pharmacological and nonpharmacological modulation in treating NAFLD.

## 1. Introduction

Chronic liver diseases associated with deranged lipid metabolism constitute a major part of the metabolic syndrome [1]. Nonalcoholic fatty liver disease (NAFLD), also known as metabolic associated fatty liver disease (MAFLD) [2], has become a global epidemic and is associated with other metabolic disorders, such as diabetes and obesity [3]. Given the paucity of treatment options available for NAFLD, there has been a surge in understanding the molecular mechanism of this disease [3]. Studies show that “lipotoxicity” is the central driving force that governs the progression of NAFLD from being a state of benign lipid accumulation to a state of lipid-induced liver damage and inflammation termed non-alcoholic steatohepatitis “NASH” [4]. Lipotoxicity at a cellular level involves perturbation in the function of several intracellular organelles, namely, mitochondria, endoplasmic reticulum (ER), and lysosomes, induced by nonesterified fatty acids or free fatty acids (FFAs), particularly those derived from saturated fats [5]. Autophagy is a cellular cytoprotective process, which is not only affected by lipids but also acts to mitigate the effect of lipotoxicity in animal models of NAFLD [6]. Autophagy inducers have shown to limit the abundance of FFAs by increasing their sequestration into triglycerides (TAGs), increasing their oxidation, and preventing the damage of intracellular organelles [7]. This review covers autophagy–lipid crosstalk and its implication in countering lipotoxicity.

## 2. Lipotoxicity: A Mechanism of Lipid-Induced Cellular Injury

The deleterious effects of lipid species such as saturated free fatty acids (SFAs) and cholesterol in nonadipose organs, termed “lipotoxicity,” was first described in 1994 by Lee et al. [8]. Lipotoxicity is a key factor that drives the progression of NAFLD [9], which is associated with type II diabetes and cardiovascular complications [4]. Given the increased global prevalence of NAFLD, understanding the mechanism of progression and developing antilipotoxic agents have gained immense attention [10]. At a molecular level, lipotoxicity is characterized by the accretion of toxic lipids in hepatic cells, leading to organelle dysfunction, cellular stress, and eventually apoptosis, which is referred to as “lipoapoptosis” [11].

Accumulation of lipids within the liver, which causes lipotoxicity, comes through five different sources [12]. Firstly, in the state of insulin resistance (IR), lipolysis in the adipose tissue is upregulated, thereby releasing FFAs in the systemic circulation. The released FFAs are taken up by the hepatocytes and stored as lipid droplets. Fatty acid transport proteins (FATPs), caveolins, and fatty acid translocase (FAT/CD36) are the main hepatic plasma membrane proteins that are involved in hepatic fatty acid uptake and contribute to the intrahepatic lipid pool. Among the FATP members, the genetic loss of FATP2 and FATP5 results in a decrease in hepatic steatosis in a diet-induced obesity mouse model [13]. Similarly, in NAFLD patients, the expression of FAT/CD36 has been shown to be upregulated and shows correlation with liver injury [14]. Moreover, the hepatocyte-specific ablation of *FAT/CD36* prevented fat accumulation in the liver and repressed inflammation [15]. Secondly, hyperinsulinemia in response to IR in hepatocytes drives the synthesis of fatty acids within the hepatocytes via a process known as “de novo lipogenesis.” This process is regulated by two transcriptional factors: sterol regulatory element-binding protein 1c (SREBP1c) and carbohydrate response element binding protein (ChREBP), which are activated in response to glucose/fructose-induced insulin release. Thirdly, hepatic lipid uptake from fat-rich food also leads to hepatic lipid accumulation. Fourthly, the impaired peroxisomal and mitochondrial oxidation of fat may result in overt hepatic lipid accumulation. Lastly, the inefficiency of hepatic cells to sequester FFAs in TAGs and secrete them as VLDL particles into the circulation also leads to increased hepatic lipotoxicity [12].

It is important to note that not all lipids cause lipotoxicity in the liver [16]. Previous studies have shown that the assembly of TAGs within the lipid droplets is often an adaptive response to counter FFAs’ influx in the liver [17]. In concordance, the loss of genes involved in TAG synthesis results in extensive damage in the liver, characterized by NASH [17]. Of note, SFAs have been shown to induce lipotoxicity but not unsaturated fatty acids, which may be attributed to their differential ability to get incorporated into TAGs [16]. Additionally, given their limited incorporation in TAGs, the free SFAs act as a precursor for these two highly bioactive lipotoxic lipid species, ceramides [18]. In line with this, mice fed with a diet rich in SFAs exhibit higher levels of ceramide than those fed with a diet rich in unsaturated fatty acids [4]. The increased levels of intrahepatic ceramide are associated with the repression of insulin signaling, leading to IR as well as increased hepatocyte damage [4]. Together with ceramide, another fatty acid metabolite known as diacylglycerol (DAG) has also been implicated in the development of IR in hepatocytes [19]. The link between hepatic DAG accumulation and hepatic IR could be attributed to the activation of PKCε, which was the predominant PKC isoform activated in the liver following fat feeding [19]. Similar to the SFAs, cholesterol exhibits lipotoxic effects in the liver, and in preclinical animal models cholesterol accumulation mediates the progression from benign steatosis to NASH [20]. Mechanistically, free cholesterol causes changes in membrane dynamics, resulting in mitochondrial and lysosomal defects, which are crucial for inducing lipotoxicity [21]. Additionally, the uptake of free cholesterol crystals by Kupffer’s cells promotes the activation of *NLRP3* and other inflammatory patterns [22]. Furthermore, free cholesterol also increases the sensitivity of hepatic stellate cells (HSCs) to transformative growth factor (TGF)-β signaling, thereby promoting fibrosis in NASH [23]. Oxidized derivatives of cholesterol, known as oxysterols, can also promote liver injury, causing mitochondrial damage within hepatocytes [20].

Organelle damage, particularly involving endoplasmic reticulum (ER) and mitochondria, is central to understanding the cellular manifestation of lipotoxicity [4]. In cases of ER, the accumulation of lipid species such as cholesterol and ceramides leads to the activation of endoplasmic reticulum (ER) stress mediators, such as activating transcription factor 4 (ATF4) and C/EBP homologous protein (CHOP), which initiate the pro-apoptotic signaling in hepatocytes [4]. Additionally, ER stress–activated transcription factors such as X-box binding protein 1 spliced (XBP-1s) also lead to the upregulation of de novo lipogenesis within hepatocytes, which further aggravates lipotoxicity [24]. Similarly, SFA-induced mitochondrial damage leads to increased oxidative stress and activation of stress kinases, including JNK and p38MAPK [25]. Besides the JNK pathway, the activation of Bcl-2–associated X protein (Bax), a pro-apoptotic member of the Blc-2 family, which is linked to the intrinsic apoptosis pathway activation, has been observed in human hepatocytes chronically treated with palmitate [26] and in high fat–fed rats [27]. Additionally, the increased rate of β-oxidation in mitochondria, in response to FFA influx, also contributes to oxidative stress within hepatocytes [25].

Toll-like receptors (TLRs) are innate immune cell-surface receptors that can be activated by FFAs [28]. Studies show that saturated FFAs participate in the activation of TLRs, resulting in the activation of inflammasome-mediated IL-1β production [29].

## 3. Autophagy and Its Modulation by Lipids

Autophagy is cellular defense mechanism that helps to counter threats from both outside and inside the cells [30]. Its role is to degrade pathogens as well as to increase the turnover of intracellular macromolecules and organelles in order to prevent cellular damage and promote cellular rejuvenation [30]. There are several types of autophagy, including macroautophagy, chaperone-mediated autophagy (CMA), and microautophagy [30]. Mechanistically, the most commonly studied, macroautophagy, involves the formation of a double membranous structure that engulfs the cellular cargo containing damaged organelles, protein aggregates, and other macromolecules, including lipids and their fusion in another intracellular organelle known as lysosomes [30]. This fused structure, known as the ‘autolysosome,’ degrades the content within the autophagosomes by the action of several pH-sensitive lysosomal hydrolases [30]. There are a number of proteins involved in the formation of autophagosomes, the recognition of a specific cargo, the fusion of autophagosomes with lysosomes, and lastly the activity of lysosomes [30]. The first set of proteins that helps in the initiation of autophagosome formation includes Unc-51–like autophagy activating kinase 1(ULK1), FIP200 and ATG13, followed by another set of *ATG* family genes including PI3K complex (*BECN1*, *VPS34*, *VPS15*, and *ATG14L*), ATG9 vesicle (*ATG9A*, *ATG2*, *WIPI1/2*), ATG12-conjugation system (*ATG7*, *ATG10*, *ATG12*, *ATG5*, and *ATG16L*), and LC3-conjugation system (*ATG4*, *ATG7*, *ATG3*, *ATG12-Atg5*, *MAP1LC3B*). The recognition of cellular cargo involves another set of proteins, such as SQSTM1, which helps in bringing the destined target within the autophagosomes [30]. The fusion of the autophagosome containing cellular cargo with the lysosomes results from the action of proteins such as Rubicon, STX17, Rab7 and VAMP8. Finally, the lysosomal membrane–associated proteins such as LAMP1, LAMP2 and V-ATPases regulate lysosomal activity [30]. Autophagy is highly sensitive to regulation by intracellular nutrient and energy levels, as well as by extracellular hormones [31]. In an anabolic state, under sufficient nutrient availability for an organism, insulin production inhibits autophagy via mammalian target of rapamycin complex 1(MTORC1) activation [31]. In contrast, under low-energy state, AMP-activated protein kinase (AMPK) activates autophagy by inhibiting MTORC1 activity [31]. Additionally, the transcriptional regulation of autophagy is also governed by several nuclear hormone receptors and via transcription factor EB (TFEB), which coordinately regulate the transcription of both autophagy and lysosomal genes [31,32].

Interestingly, lipids alter cellular autophagy, playing an important role mediating lipotoxic injury in NAFLD [33]. In fact, autophagy has been shown to be impaired in animal and human NAFLD and NASH [34].

However, the pleiotropic effect of different lipid species on autophagy is seen within the cells and is regulated by their type as well as by the exposure dose and time. In this regard, phosphoinositide (PIP3), which is a class of phospholipids derived from phosphatidylinositol, inhibits autophagy via regulation of MTORC1 activity [33]. Additionally, PIP3 also contributes toward the formation of autophagosomal membrane and mediates selected cargo capture via its interaction with autophagy-linked FYVE protein (Alfy), a nuclear scaffold protein with a FYVE domain that binds PIP3 [35]. In contrast, a metabolite of membrane phospholipid, phosphatidic acid and its derivative diacylglycerol, serves as an inhibitor of mTORC1 and positively regulates hepatic autophagy [36]. In addition to phospholipids, sphingolipids, including both ceramides and sphingosine-1-phosphate, induce autophagy via MTORC1-dependent or -independent pathways [37].

SFAs, such as palmitic acid (PA), which is a potent lipotoxic mediator, have been shown to both induce and block autophagy depending on their concentration and exposure time. Its pro-autophagic action is mediated via the activation of protein kinase C (PKC) and JNK pathways in hepatocytes [38,39]. However, the chronic exposure of PA leads to defects in autophagosomal–lysosomal fusion as well as lysosomal acidification [40]. Recently, PA was shown to impair hepatic autophagy via suppression of immune surveillance protein DDX58/Rig-1 (DExD/H box helicase 58) [41] and by activating the STING-MTORC1 pathway, which may explain the autophagy inhibition seen in advanced NAFLD [42]. Additionally, lipids and their derivatives also act as ligands for nuclear receptors such as PPARs and LXRs, which are known to transcriptionally modulate autophagy [43].

## 4. Autophagy Induction Mitigates Hepatic Lipotoxicity

Given its cytoprotective nature, autophagy induction has been tested as a possible strategy to counter lipotoxicity in hepatocytes. Given below are several aspects of autophagy-mediated antilipotoxic effects observed in cells and preclinical animal models:

### 4.1. Autophagy-Assisted TAG Assembly

As described earlier, TAG assembly into lipid droplets serves as an adaptive response to sequester FFAs within the hepatocytes [17]. This process serves to limit the lipotoxicity induced by FFAs [17] by reducing intracellular FFA and assisting lipid export out of the liver via very-low-density lipoprotein (VLDL). There are several reports that provide evidence of a prominent role of autophagy and autophagy proteins in TAG assembly and lipid droplet biogenesis [44] (Figure 1). Specifically, some of the earlier studies demonstrated that autophagy proteins MAP1LC3 [45], ATG7 [46] and FIP200 [47] are involved in the TAG assembly, and there was an observed reduction in hepatic lipid droplet number in knockout mice lacking these autophagic components. Recently, ULK1 was also shown to limit SFA-induced lipotoxicity in hepatocytes by regulating the nuclear shuttling of a nuclear corepressor, NCOR1, which leads to increased activation of LXRs and transcription of *SCD1* [48]. The activity of SCD1 leads to the conversion of SFAs into unsaturated fatty acids, which are incorporated more easily into TAGs in the hepatocytes [48,49]. Similar effects were also observed with other autophagy genes that were dependent on NCOR1 nuclear activity [50]. Additionally, the upregulation of the calcium-binding protein S100A11 augments FOXO1-mediated autophagy to increase TAG formation within the hepatocytes [51].

### 4.2. Lipophagy and Lysosomal Lipolysis

Lipophagy is a type of macroautophagic process that involves the ingestion of TAGs within the autophagosomes and their digestion within the autolysosomal compartment [7]. Although the esterification of FFAs to form TAGs is a way to reduce lipotoxicity, the prolonged storage of TAGs may cause them to become a source of lipotoxicity via lipid peroxidation [17]. Additionally, lysophosphatidylcholine (LPC), which is a component of lipid droplet–enveloped monolayers and VLDL, is an important phospholipid mediator of SFA-induced lipotoxicity in NASH [52]. Therefore, the reduction of cytosolic stores of TAGs within the hepatocytes by lipophagy is required to prevent the lipid buildup within the hepatocytes, which may lead to cellular injury [53]. Furthermore, the release of fatty acids from the action of lysosomal lipases helps to increase the mitochondrial β-oxidation and thus provide energy to hepatic cells [53]. Several pharmacological and nonpharmacological treatments have been shown to upregulate lipophagy in preclinical NAFLD models and reduce hepatic steatosis [53]. Several hormones and natural compounds/drugs such as thyroid hormone [54,55,56,57], caffeine [58], (-)-Epigallocatechin-3-gallate [59], and calcium channel blockers [60] have been suggested as novel therapeutic approaches to manage NAFLD via lipophagy. Similarly, exercise [61], calorie restriction [62], and time-restricted feeding [63] have been suggested to reduce hepatic steatosis by inducing lipophagy. Recently, the role of TFEB has been implicated in the regulation of hepatic lipophagy, and it is being investigated to see if TFEB activators may be beneficial in countering NAFLD through lipophagy induction [64]. Although the link between NAFLD and lipophagy/autophagy has been explored extensively in preclinical studies, human data are still scarce and inconclusive. Recently, Lin et al. [65] found a positive association between a variant in the lipophagy-associated gene *IRGM* and the likelihood of developing childhood NAFLD [65]. These initial studies thus provide groundwork for targeting lipophagy for countering lipotoxicity associated with NAFLD/NASH (Figure 1).

### 4.3. Protective Effect of Autophagy in Relieving Lipotoxicity-Induced Oxidative Stress

SFAs, such as PAs, are known to impair mitochondrial energetics, resulting in reduced ATP production followed by accelerated mitochondrial reactive oxygen species (ROS) production [66]. In this context, mitochondrial autophagy, known as “mitophagy,” is a process of mitochondrial pruning, which prevents the induction of lipoapoptosis in response to oxidative stress [67,68]. Defective mitophagy has been demonstrated in both human NASH and high fat diet (HFD)-induced in vivo mouse models, as well as in cultured hepatocytes treated with SFAs, associating with oxidative stress and lipoapoptosis [68]. Several mechanisms have been implicated in the regulation of mitophagy in NAFLD [69]. The expression of Acyl-CoA:lysocardiolipin acyltransferase-1 (ALCAT1) was upregulated in an HFD-induced NAFLD mouse model, and its genetic ablation can restore mitophagy in isolated hepatocytes, improving mitochondrial architecture and mtDNA fidelity and preventing the onset of hepatic steatosis in mice [70]. Furthermore, the pharmacological activation of PINK1/Parkin-dependent mitophagy by the plant flavonol quercetin alleviates HFD-induced hepatic steatosis [71]. Additionally, the overexpression of sirtuin 3 (SIRT3) activates mitophagy and rescues hepatic cells from PA-induced oxidative stress [72]. Thyroid hormone and its mimetics, which have shown promising results in human NAFLD, are known to activate hepatic autophagy via the ROS-AMPK-ULK1 pathway in human hepatic cells to limit oxidative stress [73,74] (Figure 1).

Apart from mitophagy induction, autophagy also protects against lipotoxicity-induced oxidative stress via degradation of kelch-like ECH–associated protein 1(KEAP1), which results in nuclear factor erythroid 2-like 2 (NRF2/NFE2L2)–mediated transcription of antioxidant genes [75]. In a recent study by Park et al., the authors found that autophagic protein ULK1 mitigates hepatic lipotoxicity through the activation of NRF2 [75]. KEAP1, under basal state, is a negative regulator of NRF2 activity by forming a NRF2-CUL3-RBX1 complex, leading to NRF2 degradation [76]. However, in the absence of KEAP1, NRF2 is stabilized, and its nuclear translocation activates the transcription of its target genes, including *NQO1* (NAD[P]H quinone dehydrogenase 1), *GSTA1* (glutathione S-transferase α 1), and *HMOX1/HO-1* (heme oxygenase 1) [76]. SQSTM1/p62, which is an autophagy receptor protein, relieves KEAP1-mediated NRF2 inhibition by specific binding of SQSTM1 to KEAP1, resulting in KEAP1 autophagic turnover, the stabilization of NRF2 [76]. In this context, ULK1 enhances the interactions between SQSTM1 and KEAP1 in the hepatocytes, which causes autophagic KEAP1 degradation [75,77]. Increased NRF2 levels are responsible for the induction of several antioxidant genes, which protect hepatic cells against lipid-induced oxidative stress (Figure 1).

### 4.4. Autophagy and Lipotoxicity-Associated ER Stress

The ER is a cellular hub for de novo protein synthesis and folding, which is critical to serve cellular function [78]. The perturbations in ER function interfere with protein folding in the ER, leading to proteotoxic ER stress, and hence igniting unfolded protein response (UPR), which affects several aspects of hepatic lipid metabolism [78]. Canonically, the UPR is activated via the luminal domains of three principal transmembrane sensors: inositol-requiring enzyme (IRE)-1α, protein kinase RNA-like ER kinase (PERK), and activating transcription factor (ATF)-6α [78].

SFAs including PA lead to ER stress by increasing the accumulation of di-saturated glycerolipids in the ER, which trigger sustained IRE1α and PERK activation [79], as well as by increased biosynthesis of saturated phospholipids, contributing to palmitate-induced lipotoxic ER stress [80]. At a mechanistic level, PA-induced hepatocyte lipoapoptosis occurs due to persistent UPR activation, resulting in the activation of JNK- and CHOP-mediated upregulation of proapoptotic protein, such as p53 upregulated modulator of apoptosis (PUMA) [81]. Similar to PA, other lipid species, such as lysophosphatidylcholine (LPC) [82] and ceramide [18], are also potent inducers of hepatic ER stress, which may lead to both IR and lipoapoptosis.

The misfolded proteins that cannot be repaired are eliminated from the cell through the specialized processes, ER-associated protein degradation (ERAD) and autophagy [83]. Therefore, autophagy plays a very important role in relieving ER stress induced by lipids by directly degrading misfolded proteins [83] (Figure 1). In a study using autophagy-deficient mice, HFD feeding was associated with increased hepatic ER stress and IR [84]. Intriguingly, the rescue experiments using overexpression vectors demonstrated that autophagy induction significantly rescued lipid-induced ER stress in the mouse liver along with recovery in hepatic insulin sensitivity [84].

More recently, ER-to-lysosomal-associated degradative (ERLAD) pathways describe a subset of processes that involve targeting of proteins in the ER lumen/membrane or the ER membrane itself for lysosomal degradation [85]. This involves the direct engulfment of a part of the ER by autophagosomes, and its degradation within lysosomes is called “ER-phagy” [86]. While still a less-understood process, ER-phagy may indeed play a crucial role in NAFLD pathogenesis. A role for ERLAD in NAFLD/NASH pathogenesis was highlighted by RNA sequencing data from groups that compared NASH with healthy controls. In this study, numerous ER-phagy receptors, such as ATL3, SEC62, and RTN3, were differentially regulated [83]. These data point toward ER-phagy playing an essential role during NASH and underscore its importance as a possible novel strategy for NASH treatment (Figure 1).

## 5. Pharmacological and Nonpharmacological Inducers of Autophagy and Their Regulation of Hepatic Lipotoxicity

*Pharmacological inducers of hepatic autophagy:* Several natural and synthetic molecules that have shown efficacy in inducing autophagy in cell culture and preclinical animal models of lipotoxicity are described below:

*FDA-approved drugs:* To circumvent costly and lengthy drug discovery processes, including safety assessment, dosing, and other pharmacokinetic and -dynamic characterizations, a strategy known as repurposing (or repositioning) of approved drugs has become an attractive choice to discover pro-autophagy drugs.

Some of the well-described drugs that also show pro-autophagic activity in this class are metformin [87], rapamycin [88], statins [89], pioglitazone [90], fibrates [91], canagliflozin [92], verapamil [93], and carbamazepine [94].

*Targeted/Investigational drugs:* These are molecules that specifically target a known protein involved in the autophagy pathway. One such molecule is Tat-beclin 1 (Tat-BECN1), a peptide known to stimulate autophagy through mobilization of endogenous Beclin 1 [95]. Other molecules in this class of drugs are TFEB activators [96] and ULK1 activators [97], but their role in regulating hepatic lipotoxicity remains to be investigated.

*Nutritional supplements:* Several commonly used natural products and active ingredients found in many Chinese and Indian traditional medicines have shown strong autophagy-promoting activity [98]. First in this class of compounds are caffeine and epigallocatechin gallate (EGCG), which are present in coffee and tea and are the largest consumed beverages in the world. Caffeine induces hepatic autophagy via inhibition of MTORC1 [58], whereas EGCG is an AMPK activator [59]. Resveratrol, a natural polyphenol, has been reported to improve complications associated with NAFLD via its ability to induce SIRT1-mediated autophagy [99]. Trehalose, a naturally occurring disaccharide present in plants, bacteria, fungi, insects, and certain types of shrimp, is a known inducer of autophagy and exhibits its antisteatosis action via an MTORC1 independent mechanism [100]. Ginsenoside Rb2, one of the major ginsenosides in Panax ginseng, also exhibit anti-NAFLD action in db/db mice, HepG2 cells and primary mouse hepatocytes via promoting AMPK/SIRT1-driven autophagy [101]. Additionally, akebia saponin D (ASD), extracts from *Akebia quinata* [102], and capsaicin, an extract of *Capsicum annuum* [103], a common dietary supplement, have been shown to exert beneficial effects on NAFLD via autophagy induction. Spermidine, a natural polyamine and health supplement, is a potent autophagy stimulator and counters lipid-induced liver damage via autophagy enhancement [104]. ω-3 fatty acids present in fish oil, flaxseeds, and walnuts prevent FFA-induced lipotoxicity through induction of autophagy in NAFLD [105]. Similarly, widely used components in Indian traditional medicine, curcumin and its derivatives have shown protective effects in animal models of NAFLD by an autophagy-mediated pathway [106].

*Hormones and Vitamins*: Endocrine hormones are pivotal regulators of cellular metabolism, and their deregulation has been associated with the development of lipid-associated metabolic disease, such as NAFLD. In line with this notion, the exogenous administration of some hormones, hormonal derivatives or their mimetics has shown efficacy in reducing lipotoxicity in animal models of NAFLD [31]. Thyroid hormones and their metabolites exhibit potent pro-autophagy action in vivo, mitigating lipotoxicity and hepatic lipid accumulation in hepatic cells [54,55,56,57,73]. In fact, many of the liver-specific thyroid hormone analogues are currently under human trial for NASH treatment [107,108]. Similarly, other endocrine hormones, including epinephrine [109], glucagon-like peptide-1(GLP-1) [110], ghrelin [111], and FGF21 [112], demonstrate anti-steatotic action coupled with autophagy induction in hepatic cells. Besides hormones, vitamins also induce hepatic autophagy and alleviate lipotoxicity in NAFLD preclinical models [113]. Notably, vitamin D or 1,25(OH)2 D3 diminished HFD-induced liver damage and steatosis, which was accompanied by autophagy and upregulation of ATG16L1 expression [114]. Similarly, the supplementation of vitamin B12 and folate recently has been shown to promote autophagic flux in hepatocytes, thereby reducing FFA-induced liver injury [115].

*Nonpharmacological inducers of hepatic autophagy:* Since there are no approved drugs prescribed for NAFLD/NASH in humans, exercise and lifestyle changes still remain the cornerstone to counter NAFLD-associated lipotoxicity. Several new studies highlighting the regulation of hepatic autophagy by physical activity and dietary interventions are outlined below:

*Exercise:* Exercise or physical activity has shown promising results in clinical studies as well as in experimental models of obesity and NAFLD management [116]. The beneficial effects of exercise in NAFLD have been attributed to a potent stimulation of autophagy during physical activity [61,117,118,119,120,121,122,123,124,125]. Specifically, exercise has been directly linked to lipophagy-mediated TAG turnover [119] as well as mitophagy induction to counter FFA-induced mitochondrial damage [123].

*Calorie restriction, Chrono-nutrition, and dietary interventions:* Calorie restriction, independent of the macronutrient composition, is associated with NAFLD improvement and reduction of hepatic fat [126,127,128]. Similarly, low-carb diets have shown beneficial effects in reducing hepatic steatosis in humans [126]. Interestingly, using preclinical animal models, both calorie restriction and time-restricted feeding have been shown to induce hepatic autophagy [63,129,130].

## 6. Conclusions

Based on the urgent clinical need to treat metabolic liver diseases such as NAFLD, understanding and targeting lipotoxicity have become a cornerstone of NAFLD research. Autophagy has been shown to be efficient in preventing several aspects of lipotoxicity in preclinical animal models. Further studies in humans are needed to realize its translational potential. As autophagy induction can be achieved by several nutraceuticals and nonpharmacological approaches, further studies in this direction may broaden the horizon of treatment strategies for NAFLD.

## Figures and Tables

**Figure 1 genes-14-00553-f001:**
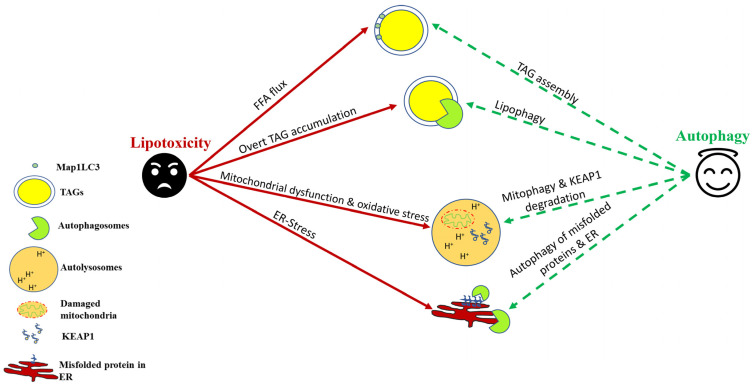
Autophagic induction in hepatocytes confers protection against lipotoxicity. As a cellular defense against nonesterified free fatty acid (FFA) flux, autophagy helps in their sequestration into triglycerides (TAGs). Lipophagy helps in TAG lipolysis, which helps in β-oxidation of stored lipids. Autophagy–lysosomal degradation of damaged mitochondria and KEAP1 helps to counter oxidative stress induced by saturated fats and their derivatives. Autophagy also helps to mitigate ER stress by degrading misfolded proteins and, in certain cases, a part of ER itself through a process termed “ER-phagy.”

## Data Availability

Not Applicable.

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
