# Peer review of "Autophagy: A Cellular Guardian against Hepatic Lipotoxicity"

_genes, 2023, doi:10.3390/genes14030553_

Round 1
Reviewer 1 Report
The manuscript by Sinha represents an informative and focused review on the protective role of Autophagy in hepatic lipotoxicity. This review might be an interesting topic and gives the reader some important insight without too much reading. Below is my critique for the authors to further improve their work:
1. The author should introduce how the concept of Lipotoxicity was discovered in the introduction (Ref. Proc Natl Acad Sci USA. 91:10878-10882. 1994).
2. Page 2, Line 69-71; - two highly bioactive lipotoxic lipid -, What is two in this sentence.
3. Page 3, Line111-112; The author should describe more detail as following; ATG family genes including PI3K complex (BECN1, VPS34, VPS15, and ATG14L), ATG9 vesicle (ATG9A, ATG2, WIPI1/2), ATG12-conjugation system (ATG7, ATG10, ATG12, ATG5, and ATG16L), and LC3-conjugation system (ATG4, ATG7, ATG3, ATG12-Atg5, MAP1LC3B).
4. Regarding the description of TFEB, the general function in autophagy pathway should be mentioned before the description on page 5, line 195, as referred to Science. 33:1429-1433, 2011.
5. In Figure 1, The arrow shadows should be removed. Autophagosome/isolation membrane should be represented as a double membrane structure.
Author Response
We are thankful to the reviewer for his/her valuable comments. As suggested, we have made changes accordingly in the revised manuscript.
Reviewer: 1
Comments and Suggestions for Authors
The manuscript by Sinha represents an informative and focused review on the protective role of Autophagy in hepatic lipotoxicity. This review might be an interesting topic and gives the reader some important insight without too much reading. Below is my critique for the authors to further improve their work:
Response: I am thankful to the reviewer for his/her comments. The revised manuscript has addressed all the concerns raised.
- The author should introduce how the concept of Lipotoxicity was discovered in the introduction (Ref. Proc Natl Acad Sci USA. 91:10878-10882. 1994). Response: This Reference has been know incorporated as pointed out.
- Page 2, Line 69-71; - two highly bioactive lipotoxic lipid -, What is two in this sentence. Response: The correction has been done.
- Page 3, Line111-112; The author should describe more detail as following; ATG family genes including PI3K complex (BECN1, VPS34, VPS15, and ATG14L), ATG9 vesicle (ATG9A, ATG2, WIPI1/2), ATG12-conjugation system (ATG7, ATG10, ATG12, ATG5, and ATG16L), and LC3-conjugation system (ATG4, ATG7, ATG3, ATG12-Atg5, MAP1LC3B). Response: The required amendments have been done.
- Regarding the description of TFEB, the general function in autophagy pathway should be mentioned before the description on page 5, line 195, as referred to Science. 33:1429-1433, 2011. Response: This Reference has been know incorporated as pointed out.
- In Figure 1, The arrow shadows should be removed. Autophagosome/isolation membrane should be represented as a double membrane structure. Response: The required amendments have been done.
Reviewer 2 Report
Autophagy’s functions in non-alcoholic fatty liver disease by lipid metabolism. The design is novel and there are several aspects must be revised.
1: References of 8,38,42,44,48,53,65 and 72 changed.
2: Keywords: keyword 1; keyword 2; keyword 3 (List three to ten pertinent keywords specific to the 20 article yet reasonably common within the subject discipline. Details are not clearly.
3: Authors contributions:The following statements should be used “Conceptual- 286 ization, X.X. and Y.Y.; methodology, X.X.; software, X.X.; validation, X.X., Y.Y. and Z.Z.; formal 287 analysis, X.X.; investigation, X.X.; resources, X.X.; data curation, X.X.; writing—original draft prep- 288 aration, X.X.; writing—review and editing, X.X.; visualization, X.X.; supervision, X.X.; project ad- 289 ministration, X.X.; funding acquisition, Y.Y. All authors have read and agreed to the published ver- 290 sion of the manuscript.”. Details ?
Author Response
We are thankful to the reviewer and the revised manuscript has been amended as requested.
Reviewer 2:
Autophagy’s functions in non-alcoholic fatty liver disease by lipid metabolism. The design is novel and there are several aspects must be revised.
1: References of 8,38,42,44,48,53,65 and 72 changed.
Response: I am not sure I understood this point by the reviewer, however I have checked that the references have been correctly cited in the text.
2: Keywords: keyword 1; keyword 2; keyword 3 (List three to ten pertinent keywords specific to the 20 article yet reasonably common within the subject discipline. Details are not clearly.
Response: The required amendments have been done.
3: Authors contributions:The following statements should be used “Conceptual- 286 ization, X.X. and Y.Y.; methodology, X.X.; software, X.X.; validation, X.X., Y.Y. and Z.Z.; formal 287 analysis, X.X.; investigation, X.X.; resources, X.X.; data curation, X.X.; writing—original draft prep- 288 aration, X.X.; writing—review and editing, X.X.; visualization, X.X.; supervision, X.X.; project ad- 289 ministration, X.X.; funding acquisition, Y.Y. All authors have read and agreed to the published ver- 290 sion of the manuscript.”. Details ?
Response: The required amendments have been done.
Reviewer 3 Report
REVIEW – Autophagy: A cellular guardian against hepatic lipotoxicity
The authors have submitted a review manuscript to Genes (manuscript ID: 2133785) which synthesis the current scientific literature on autophagy processes to counteract lipotoxicity as a cause of hepatic metabolic liver disease. Please consider the below comments and in-text corrections for minor revision of the manuscript.
The submitted review article consist of an unstructured abstract, no keywords (missing), five subsections as well as 75 references, 1 figure with caption and part of the required statements (missing contribution and acknowledgments). It is strongly recommended to adhere to the publication guidelines of the journal and to respect the structure requested as provided by the journal.
The author present a comprehensive review on the currently available literature in a well-structured and well-written article. An overview of the autophagy-lipotoxicity interaction is relevant for the readers of genes, because non-alcoholic fatty liver disease is an entity with growing importance due to increasing prevalence of the condition. Thus, a profound understanding of the molecular mechanisms and the consequences for human physiology and pathology are of significant importance.
The present review adequately introduces the need for research on autophagy and lipotoxicity and details the mechanisms of lipotoxicity and autophagy and how the latter can mitigate the toxic effects.
While the review discusses potential treatment targets, this could be further elaborated. In addition, it may be interesting to understand the consequences of lipotoxicity in other organs, especially the brain. It is understandable that this is not discussed in the present review because it would go beyond the scope of the present topic.
Overall, highly interesting and well made review which will hopefully result in a substantial number of experimental studies.
In-text corrections
L65: remove “is”
L125-126: “...and this plays an important link between hepatic autophagy the severity of lipotoxicity in metabolic liver diseases such as MAFLD[22].” This sentence should be rephrased.
L175-176: “Autophagy also helps to mitigate ER-stress by degrading misfolded proteins and certain cases a part of ER itself termed as ER-phagy”. This sentence should be rephrased.
L182-183: “Additionally, lysophosphatidylcholine (LPC), which is a lipid droplet envelop monolayers, and VLDL is an important phospholipid mediator of SFA induced lipotoxicity in NASH[41].” This sentence should be rephrased.
L191: remove “that”
L199-202: “Recently, Lin et al. [54] genotyped 832 obese children (aged 6–18 years) of East Asian de-199 scent and found that a variant in the lipophagy associated gene IRGM, diagnosed using an
ultrasonography based scoring pattern described by approximately 2-fold, increased the chance to develop NAFLD [54].” This sentence should be rephrased.
L241: Change “folding critical” to ” folding is critical”.
L254: There is a hyperlink behind “ER” which should be removed.
L272-274: Change “This data points towards ER-phagy playing an essential role during NASH and underscore its importance as a possible novel treatment strategy for NASH 273 treatment (Figure 1).”
To “This data points towards ER-phagy playing an essential role during NASH and underscores its importance as a possible novel strategy for NASH treatment (Figure 1).”
L278-L280: Change “Autophagy has been shown to be significantly efficient in preventing several aspects of lipotoxicity in pre-clinical animal models, and future studies in humans are needed to realize
its translational potential.” to “Autophagy has shown to be efficient in preventing several aspects of lipotoxicity in pre-clinical animal models. Further studies in humans are needed to realize its
translational potential.
Author Response
We are grateful to the reviewer for his/her critical suggestions. We have made all necessary corrections as suggested in our revised manuscript.
Reviewer 3:
The authors have submitted a review manuscript to Genes (manuscript ID: 2133785) which synthesis the current scientific literature on autophagy processes to counteract lipotoxicity as a cause of hepatic metabolic liver disease. Please consider the below comments and in-text corrections for minor revision of the manuscript.
The submitted review article consist of an unstructured abstract, no keywords (missing), five subsections as well as 75 references, 1 figure with caption and part of the required statements (missing contribution and acknowledgments). It is strongly recommended to adhere to the publication guidelines of the journal and to respect the structure requested as provided by the journal.
The author present a comprehensive review on the currently available literature in a well-structured and well-written article. An overview of the autophagy-lipotoxicity interaction is relevant for the readers of genes, because non-alcoholic fatty liver disease is an entity with growing importance due to increasing prevalence of the condition. Thus, a profound understanding of the molecular mechanisms and the consequences for human physiology and pathology are of significant importance.
The present review adequately introduces the need for research on autophagy and lipotoxicity and details the mechanisms of lipotoxicity and autophagy and how the latter can mitigate the toxic effects. While the review discusses potential treatment targets, this could be further elaborated. In addition, it may be interesting to understand the consequences of lipotoxicity in other organs, especially the brain. It is understandable that this is not discussed in the present review because it would go beyond the scope of the present topic.
Overall, highly interesting and well made review which will hopefully result in a substantial number of experimental studies.
Response: I am thankful to the reviewer for his/her comments. The revised manuscript has addressed all the concerns raised.
In-text corrections
L65: remove “is”
L125-126: “...and this plays an important link between hepatic autophagy the severity of lipotoxicity in metabolic liver diseases such as MAFLD[22].” This sentence should be rephrased.
L175-176: “Autophagy also helps to mitigate ER-stress by degrading misfolded proteins and certain cases a part of ER itself termed as ER-phagy”. This sentence should be rephrased.
L182-183: “Additionally, lysophosphatidylcholine (LPC), which is a lipid droplet envelop monolayers, and VLDL is an important phospholipid mediator of SFA induced lipotoxicity in NASH[41].” This sentence should be rephrased.
L191: remove “that”
L199-202: “Recently, Lin et al. [54] genotyped 832 obese children (aged 6–18 years) of East Asian de-199 scent and found that a variant in the lipophagy associated gene IRGM, diagnosed using an ultrasonography based scoring pattern described by approximately 2-fold, increased the chance to develop NAFLD [54].” This sentence should be rephrased.
L241: Change “folding critical” to ” folding is critical”.
L254: There is a hyperlink behind “ER” which should be removed.
L272-274: Change “This data points towards ER-phagy playing an essential role during NASH and underscore its importance as a possible novel treatment strategy for NASH 273 treatment (Figure 1).” To “This data points towards ER-phagy playing an essential role during NASH and underscores its importance as a possible novel strategy for NASH treatment (Figure 1).”
L278-L280: Change “Autophagy has been shown to be significantly efficient in preventing several aspects of lipotoxicity in pre-clinical animal models, and future studies in humans are needed to realize its translational potential.” to “Autophagy has shown to be efficient in preventing several aspects of lipotoxicity in pre-clinical animal models. Further studies in humans are needed to realize its translational potential.
Response: I am thankful to the reviewer pointing out these errors. All these in-text corrections have been done in the revised manuscript.